# First Molecular Evidence of *Babesia caballi* and *Theileria equi* in Imported Donkeys from Kyrgyzstan

**DOI:** 10.3390/pathogens13090713

**Published:** 2024-08-23

**Authors:** Xuanchen Wu, Jun Xu, Lixin Su, Ente Li, Suwen Wang, Sándor Hornok, Gang Liu, Yuanzhi Wang

**Affiliations:** 1NHC Key Laboratory of Prevention and Treatment of Central Asia High Incidence Diseases, School of Medicine, Shihezi University, Shihezi 832002, China; 2Urumqi Customs, Urumqi 833400, China; 3Department of Parasitology and Zoology, University of Veterinary Medicine, 1078 Budapest, Hungary; 4HUN-REN-UVMB Climate Change: New Blood-Sucking Parasites and Vector-Borne Pathogens Research Group, 1078 Budapest, Hungary

**Keywords:** Equine piroplasmosis, *RAP-1*, *EMA-1*, donkeys, Kyrgyzstan

## Abstract

Equine piroplasmosis (EP) is an important tick-borne disease of equids, caused by Theileria equi, Theileria haneyi, and Babesia caballi. Nonetheless, there has been a scarcity of systematic reports on EP parasites in donkeys in Kyrgyzstan, Central Asia. In this study, piroplasms were screened in 1900 blood samples from imported donkeys from the Osh Oblast (southwestern Kyrgyzstan) by targeting partial 18S ribosomal RNA using the polymerase chain reaction (PCR). Through molecular and phylogenetic analyses, all positive samples were sequenced to identify the species and genotypes. The results indicated the presence of both B. caballi and T. equi, with prevalence rates of 8.4% (160/1900) and 12.2% (232/1900), respectively. By amplifying part of the Erythrocyte Merozoite Antigen 1 (EMA-1) and Rhoptry-Associated Protein (RAP-1) genes, B. caballi genotype B and T. equi genotype A were identified. To the best of our knowledge, this is the first report on piroplasm infection among donkeys from Kyrgyzstan.

## 1. Introduction

Equine piroplasmosis (EP) is a tick-borne disease caused by the protozoa *Babesia caballi*, *Theileria equi* and *Theileria haneyi* that affect equids, including domestic (horses, donkeys, and mules) and wild species (Przewalski’s horses, wild donkeys, and zebras) [1,2]. Despite their biological differences, these parasites induce similar pathogenicity, sharing comparable life cycles and vector relationships [3]. Equine piroplasms are biologically transmitted by ticks of the family Ixodidae [4], particularly *Dermacentor*, *Rhipicephalus*, and *Hyalomma* spp., which are prevalent in tropical, subtropical, and temperate climate zones [5,6]. These vectors have been identified in regions such as Asia, South and Central America, Africa, Southern Europe, and parts of the southern USA [7]. In its acute form, EP can cause donkeys to exhibit symptoms like fever, listlessness, depression, pronounced thirst, eyelid edema, constipation, yellow mucous on feces, urine discoloration, and splenomegaly [8]. However, in the more common chronic form, the signs in donkeys are usually nonspecific, manifesting as weight loss, poor physical condition, and overall poor health, and they frequently coincide with other diseases (e.g., anaplasmosis), not only in horses, but also in donkeys [9,10]. Additionally, donkeys may present an asymptomatic infection compared to horses, and they tend to exhibit lower levels of parasitemia [11].

Kyrgyzstan is predominantly an agricultural nation, with over 40% of its workforce engaged in agricultural activities. The livestock sector is a significant part of the agricultural industry [12]. Certain zoonotic diseases (ZDs) such as anthrax, brucellosis, alveolar echinococcosis, cystic echinococcosis, toxoplasmosis, and non-typhoidal salmonellosis are considered to have a major socioeconomic impact in Kyrgyzstan, posing a high risk to 64% of the population living in rural areas [13].

In Kyrgyzstan, research on piroplasmosis has been limited to dogs and cattle. In particular, the prevalence rate of piroplasms was 6.23% in dogs, while *Theileria orientalis*, *Babesia bigemina*, *Theileria annulata*, and *Babesia bovis* showed infection rates of 84.3%, 47.3%, 16.6%, and 2.5% in cattle, respectively [14,15]. However, little is known about piroplasm infection in donkeys in Kyrgyzstan. To better understand the infection rates and genotypes of Piroplasmida among donkeys in the region, a total of 1900 blood samples were collected in the Osh Oblast (southwestern Kyrgyzstan), followed by molecular analyses.

## 2. Materials and Methods

### 2.1. Study Area and Collection of Blood Samples

The Republic of Kyrgyzstan, a Central Asian country, is bordered by Uzbekistan, Kazakhstan, Tajikistan, and China. It comprises seven administrative oblasts: Chuy, Jalal-Abad, Talas, Naryn, Batken, Issyk-Kul, and Osh. In the period from November to December 2023, a total of 1900 blood samples were collected from imported donkeys in the Osh Oblast (southwestern Kyrgyzstan). All the donkeys appeared to be healthy. Sampling was conducted via jugular venipuncture at the China–Kyrgyzstan border port. All samples were drawn into EDTA anticoagulant vacuum tubes and transported to the laboratory within four hours.

### 2.2. DNA Extraction and Molecular Detection of Piroplasmida

Total DNA extraction from each blood sample was carried out with a Blood Genomic DNA Extraction Kit (Vazyme Biotech Co., Ltd., Nanjing, China) according to the manufacturer’s instructions. An approximately 400 bp fragment of the *18S rRNA* gene was used for screening piroplasms in donkey DNA samples [16]. To confirm the genotypes, we further amplified and sequenced the partial Erythrocyte Merozoite Antigen 1 gene (*EMA-1*, 750 bp) of *T. equi* and an approximately 568 bp long fragment of the Rhoptry-Associated Protein (*RAP-1*) gene of *B. caballi* [17,18]. Double distilled water served as the negative control, while sequence-confirmed *Babesia* and *Theileria* DNA amplified in our laboratory were used as positive controls [19]. The primers and PCR cycling conditions in this study are shown in Appendix A.

### 2.3. Sequencing and Phylogenetic Analysis

PCR products were purified using the TIANgel Midi Purification Kit (TIANGEN Biotech Co., Ltd., Beijing, China), inserted into the pMD-18T vector (TaKaRa Biotech Co., Ltd., Dalian, China), and then sequenced. The sequencing results were compared to others deposited in GenBank with the BLASTn program (http://www.ncbi.nlm.nih.gov/BLAST/, accessed on 9 August 2024). Phylogenic trees were constructed using the Maximum Likelihood method with MEGA 7.0 software. Bootstrap values were performed with 1000 replicates.

## 3. Results

In this study, 8.42% (160/1900) of the samples were found to be infected with *B. caballi*. BLASTn analyses revealed that the sequences of the *18S rRNA* gene of *B. caballi* showed 96.57-to-100% nucleotide identities within Kyrgyzstan. On the other hand, there was a lower 91.91-to-100% nucleotide identity between *B. caballi* isolates present in Kyrgyzstan, and other *B. caballi* isolates from GenBank, originating from various parts of the world.

Interestingly, *B. caballi* isolates in this study shared 100% nucleotide identities with the corresponding pathogens from horses in Israel (MN629354 and MK288106), Senegal (MG052892), and South Africa (EU888904 and Z15104), while *Hyalomma asiaticum* corresponded with horses in China (MF120934 and MH104643), Italy (KX375824), and Bulgaria (MG972849) (shown in Figure 1).

In addition, 12.21% (232/1900) of the samples were found to be infected with *T. equi*. BLAST analyses indicated that the partial sequences of the *18S rRNA* gene of *T. equi* in this study showed 98.46–100% nucleotide identities among each other.

However, only a 90.78-to-99.76% nucleotide identity was found between *T. equi* from Kyrgyzstan, and other isolates from different parts of the world were taken from GenBank. Furthermore, the *T.equi* in this work, respectively, shared 100% nucleotide identity with the corresponding pathogens from horses in Chile (MT463613), Egypt (MN625898), Israel (MK392053 and KX227640), Turkey (MG569905), South Africa (EU888906), France (MF510478), Brazil (KY464035, PP249544, and KJ573370), the USA (JX177673), Cuba (MT463336), and Saudi Arabia (LC431545 and KJ801931).

Phylogenetic analysis revealed that *B. caballi* and *T. equi* from Kyrgyzstan clustered with genotype B and genotype A, respectively (shown in Figure 1). The phylogenetic tree of the *RAP-1* gene indicated that the sequence of *B. caballi* from Kyrgyzstan clustered with genotype B1 (shown in Figure 2). In addition, the phylogenetic tree of the *EMA-1* gene showed that the sequence of *T. equi* from Kyrgyzstan clustered in genotype group 1 (shown in Figure 3). No co-infections with piroplasms were found among the 1900 blood samples.

## 4. Discussion

This study may help to shed light on the prevalence of *Babesia* and *Theileria* species among donkeys in Kyrgyzstan and the diversity of *Babesia* and *Theileria* in Central Asia. *B*. *caballi* and *T*. *equi* were detected in these imported donkeys. Genotyping analysis indicated *B. caballi* genotype B and *T. equi* genotype A. To the best of our knowledge, this is the first report of piroplasm infection in donkeys in Kyrgyzstan.

*Theileria equi* and *B*. *caballi* are obligate intraerythrocytic parasites that infect a wide range of domestic equids (horses, donkeys, and mules), zebras (*Equus zebra*, *Equus quagga*, and *Equus grevyi*), and wild asses (*Equus africanus*, *Equus hemonius*, and *Equus kiang*) [20]. Based on genetic variations of the *18S rRNA* gene, *B*. *caballi* has been classified into types, A, B (or B1), and C (or B2), which have been identified across Asia, Europe, Africa, and South America [20]. Notably, genotype B is prevalent in South Africa, Nigeria, and Mongolia [18,20,21]. Previously, *B*. *caballi* from *Dermacentor marginatus* and *Hyalomma asiaticum* in Almaty and South Kazakhstan oblasts was identified as genotype A [19].

Moreover, *T*. *equi* has been divided into five genotypes based on the following *18S rRNA* gene sub-lineages: A, B, C, D, and E. Genotype A is widespread in most countries across all continents, except Australia [20]. *Theileia equi* isolates from domestic donkeys in Israel and the Palestinian Authority (PA) were classified as genotype D, while those from wild donkeys (*Equus africanus asinus*) in the same regions were identified as genotype A [2]. Isolates from free-roaming donkeys in Sri Lanka were identified as genotypes C and D [22]. Our earlier research in South Kazakhstan Oblast showed that *T*. *equi* isolates from *D. marginatus* belonged to genotype E [19]. In this study, *T*. *equi* genotype A was detected in donkeys in southwestern Kyrgyzstan.

In summary, the genetic diversity between *B*. *caballi* and *T*. *equi* in Kyrgyzstan and Kazakhstan highlights the genotypic complexity of Piroplasmida in the Central Asian region. This diversity is crucial for understanding the epidemiology of equine piroplasmosis and will influence the implementation of strategies for disease management and control. The variation in genotypes may be attributed to differences in geographic distribution, host species, and/or tick vectors [2]. These discoveries are pivotal for advancing the epidemiological understanding of Piroplasmida in Central Asia.

## 5. Conclusions

To our knowledge, this study represents the first molecular evidence of *B. caballi* and *T. equi* in donkeys from Kyrgyzstan, confirming the presence of *B. caballi* genotype B and *T. equi* genotype A. These discoveries are pivotal for advancing the epidemiological understanding of Piroplasmida in Central Asia.

## Figures and Tables

**Figure 1 pathogens-13-00713-f001:**
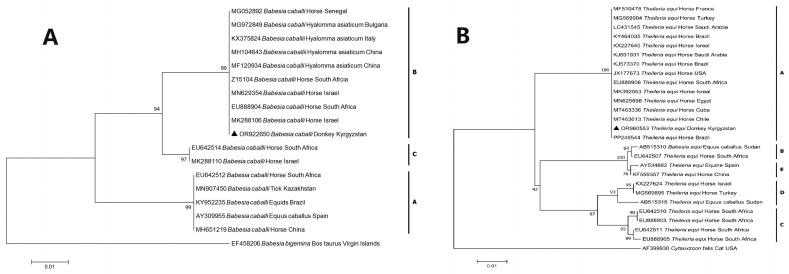
Phylogenetic analysis of piroplasms with MEGA7.0. The trees were constructed with the Maximum Likelihood method (ML; bootstrap replicates: 1000). Branch lengths correlate to the number of substitutions inferred according to the scale shown. Sequences obtained in this study are indicated by solid triangles (▲). (**A**): Analysis based on the *Babesia 18S rRNA* gene fragment; (**B**): analysis based on the *Theileria 18S rRNA* gene fragment.

**Figure 2 pathogens-13-00713-f002:**
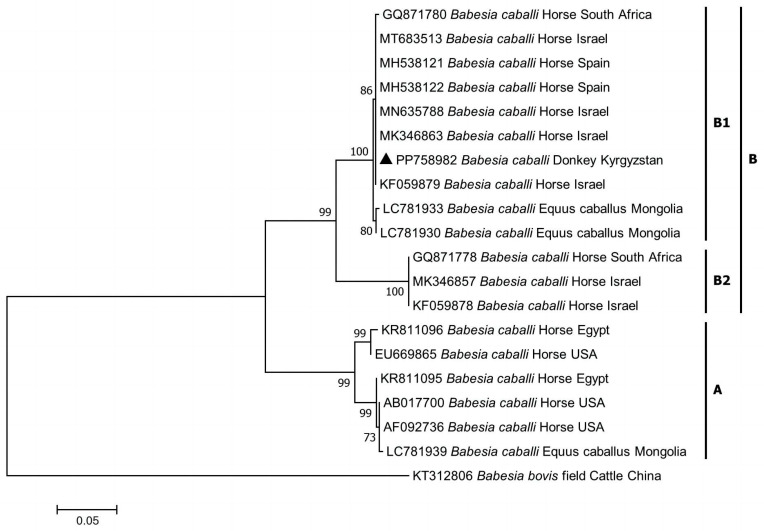
Phylogenetic analysis of piroplasms based on the *Babesia caballi RAP-1* gene fragment with MEGA7.0. The tree was constructed with the Maximum Likelihood method (ML; bootstrap replicates: 1000). Branch lengths correlate with the number of substitutions inferred according to the scale shown. The sequence obtained in this study is indicated by the solid triangle (▲).

**Figure 3 pathogens-13-00713-f003:**
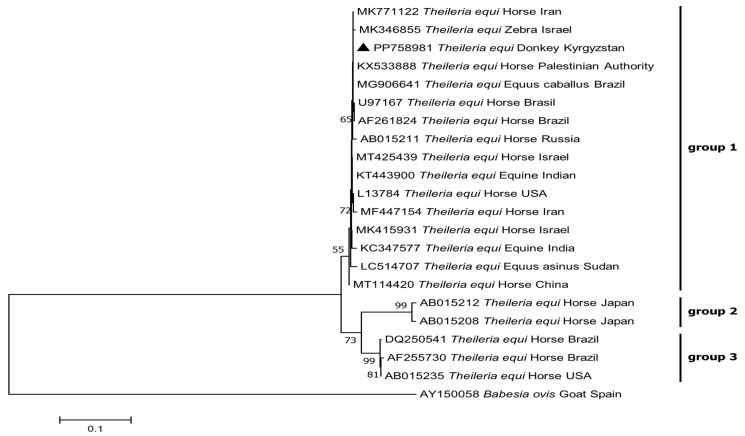
Phylogenetic analysis of piroplasms based on the *Theileria equi EMA-1* gene fragment with MEGA7.0. The tree was constructed with the Maximum Likelihood method (ML; bootstrap replicates: 1000). Branch lengths correlate with the number of substitutions inferred according to the scale shown. The sequence obtained in this study is indicated by the solid triangle (▲).

## Data Availability

The sequences from our study were deposited into GenBank (*Babesia caballi 18S rRNA*: OR922650; *Theileria equi 18SrRNA*: OR960553; *Babesia caballi rap-1*: PP758982; and *Theileria equi EMA-1*: PP758981).

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
