# Peer review of "First Molecular Evidence of Babesia caballi and Theileria equi in Imported Donkeys from Kyrgyzstan"

_pathogens, 2024, doi:10.3390/pathogens13090713_

Round 1

Reviewer 1 Report

Comments and Suggestions for Authors

Manuscript is nicely written, check the highlights corrections on the attached manuscript.

Delete (Sang et al 2021) from the first line on page 5, keep only the reference (17).

check italics for scientific names on the references. see attached file.

Reviewer 2 Report

Comments and Suggestions for Authors

Dear authors,

This paper provides important information for the analysis of piroplasmids in Kyrgyzstan. Some comments have been added to the attached PDF to improve this paper.

Comments on the Quality of English Language

Some editings need to be done before publishing.

Reviewer 3 Report

Comments and Suggestions for Authors

the manuscript entitle "First Molecular Evidence of Babesia caballi and Theileria equi in Imported Donkeys from Kyrgyzstan" is an intersting study about the epidemiological distribution of B. caballi and T. equi in Central Asia. Minor revision are required

the introduction section is fluent and well written. in the description of EP only the acute form was indicated, the presence of cronich form that more frequent in endemic are should be reported. such as the frequent concomitation of EP with other diseases (Anaplasmosis) (Journal of Equine Veterinary Science,

Volume 31, Issue 4, 2011, https://doi.org/10.1016/j.jevs.2011.02.004.) not only in horses, but also in donkey (Parasitol Res 111, 951–955 (2012). https://doi.org/10.1007/s00436-012-2854-5)

in the aim of the study the clinical status of the animals involved should be indicated.

which kind of statistical analyses was performed?

Author Response

Comments 1: in the description of EP only the acute form was indicated, the presence of chronic form that more frequent in endemic are should be reported. such as the frequent concomitation of EP with other diseases (Anaplasmosis) (Journal of Equine Veterinary Science, Volume 31, Issue 4, 2011, https://doi.org/10.1016/j.jevs.2011.02.004.) not only in horses, but also in donkey (Parasitol Res 111, 951–955 (2012). https://doi.org/10.1007/s00436-012-2854-5)

Response 1: Agree, I added the description of chronic form and I cited the two papers you gave me to emphasize this article. I marked the revisions in red in the revised manuscript. This change can be found - page 1, first paragraph of Introduction.

Comments 2: in the aim of the study the clinical status of the animals involved should be indicated.

Response 2: Thank you for pointing this out. I agree with this comment. Therefore, I have added a sentence "All the donkeys are in good health.". This change can be found - page 2, paragraph 3, and line 5.

Comments 3: which kind of statistical analyses was performed?

Response 3: No statistical analyses was used.
